# *Cannabis sativa* Extract Induces Apoptosis in Human Pancreatic 3D Cancer Models: Importance of Major Antioxidant Molecules Present Therein

**DOI:** 10.3390/molecules27041214

**Published:** 2022-02-11

**Authors:** Fathi Emhemmed, Minjie Zhao, Selvi Yorulmaz, Damien Steyer, Celine Leitao, Marion Alignan, Muriel Cerny, Alexandra Paillard, Franck Milone Delacourt, Diane Julien-David, Christian D. Muller

**Affiliations:** 1IPHC, UMR 7178 CNRS, Faculté de Pharmacie, Université de Strasbourg, 67401 Illkirch, France; fathi.emhemmed@iphc.cnrs.fr (F.E.); minjzhao@unistra.fr (M.Z.); selvi.yorulmaz@etu.unistra.fr (S.Y.); diane.julien-david@unistra.fr (D.J.-D.); 2Twistaroma, 300 Bd Sébastien Brant, 67412 Illkirch, France; damien.steyer@twistaroma.fr (D.S.); celine.leitao@twistaroma.fr (C.L.); 3Laboratoire de Chimie Agro-Industrielle, LCA, Université de Toulouse, INRAE, 31400 Toulouse, France; marion.alignan@toulouse-inp.fr (M.A.); muriel.cerny@ensiacet.fr (M.C.); 4DelleD/La Fleur, 1, rue Fleming, 49100 Angers, France; alexandra@delled.fr (A.P.); franck@delled.fr (F.M.D.)

**Keywords:** *Cannabis sativa* L., phytocannabinoids, phytochemicals, antioxidants, ROS, proapoptotic activity, caspase activation, TNF secretion, pancreatic cancer, human

## Abstract

In recent years, interest in *Cannabis sativa* L. has been rising, as legislation is moving in the right direction. This plant has been known and used for thousands of years for its many active ingredients that lead to various therapeutic effects (pain management, anti-inflammatory, antioxidant, etc.). In this report, our objective was to optimize a method for the extraction of cannabinoids from a clone of *Cannabis sativa* L. #138 resulting from an agronomic test (LaFleur, Angers, FR). Thus, we wished to identify compounds with anticancer activity on human pancreatic tumor cell lines. Three static maceration procedures, with different extraction parameters, were compared based on their median inhibitory concentration (IC_50_) values and cannabinoid extraction yield. As CBD emerged as the molecule responsible for inducing apoptosis in the human pancreatic cancer cell line, a CBD-rich cannabis strain remains attractive for therapeutic applications. Additionally, while gemcitabine, a gold standard drug in the treatment of pancreatic cancer, only triggers cell cycle arrest in G0/G1, CBD also activates the cell signaling cascade to lead to programmed cell death. Our results emphasize the potential of natural products issued from medicinal hemp for pancreatic cancer therapy, as they lead to an accumulation of intracellular superoxide ions, affect the mitochondrial membrane potential, induce G1 cell cycle arrest, and ultimately drive the pancreatic cancer cell to lethal apoptosis.

## 1. Introduction

Plants have been an important source of new pharmacologically active compounds, with many blockbuster drugs being derived directly or indirectly from plants. Alongside synthetic chemistry, as a means of discovering and making medicines, the contribution of plants to the treatment and prevention of disease remains enormous. Obviously, natural products will continue to be extremely important as sources of medicinal agents. Besides the natural products which have been found for direct medical application as medicinal entities, many others can serve as chemical models, or as models for the conception, synthesis, and semi-synthesis of new substances intended for the treatment of human diseases.

The use of cannabinoids as anti-cancer therapeutics has been extensively studied. It can be concluded that they generally exert protective and beneficiary effects, inhibiting tumor growth and progression and restoring homeostasis. Although the clinical use of cannabinoids in palliative care has been extensively documented, clinical trials on their application as anti-cancer drugs are still ongoing. As drug repurposing is significantly faster and more economical than de novo introduction of a new drug into the clinic, there is hope that the existing pharmacokinetic and safety data on the cannabinoid receptors’ (CB1R and CB2R) ligands will contribute to their successful translation into oncological healthcare [1,2]. CB1R and CB2R are members of a large family of membrane proteins called G protein-coupled receptors (GPCR). These protein complexes possess unique pharmacological and signaling properties, and their modulation might induce antitumor activity. We will here emphasize the potential of natural products issued from medicinal hemp for anti-cancer therapies, especially of pancreatic origins.

The main objectives of our work were: (i) to compare three different cannabinoid extraction methods; (ii) to identify and quantify the compounds exhibiting anticancer activity on human pancreatic; and (iii) to study the mechanism of action for apoptosis induction in a pancreatic cancer line described for drug resistance. Thus, this was a mechanistic study never before performed with human pancreatic chemo-resistant cancer organoids. Thus, a method of identification and quantification of cannabinoids in cannabis extract #138 was established using HPLC-UV. The anticancer action of cannabis extract #138 on human pancreatic tumor cell lines was analyzed by capillary flow cytometry and labeling with annexin V-FITC and propidium iodide (IP). Finally, we wanted to elucidate which mechanism the pro-apoptotic activity of our extract ultimately led to programmed cell death.

## 2. Materials and Methods

### 2.1. Material

#### 2.1.1. Chemicals and Reagents

All solvents used in this study were of analytical grade. Ethanol and acetic acid were purchased from Merck Sigma Aldrich (Darmstadt, Germany). Methanol, DMSO and ethyl acetate were purchased from VWR (Rosny-sous-Bois, France). Acetonitrile (HPLC grade), formic acid (LC-MS grade LiChropur™, 97.5–98.5%) and ammonium formate (LC-MS grade, LiChropur™, ≥99.0%) were purchased from Merck (Darmstadt, Germany), and absolute ethanol (≥99.8%, AnalaR NORMAPUR^®^ ACS, Reag. Ph. Eur.) from VWR (Rosny-sous-Bois, France). Purified water was obtained from a Merck Millipore Milli Q system (Millipore, Bedford, MA, USA), potassium persulfate, 2,2′-azo-bis(2-methylpropionamidine) dihydrochloride (AAPH), Gemcitabine and fluorescein were purchased from Merck-Sigma-Aldrich (Germany). Cannabinoid analytical standards (cannabidiol (CBD), cannabidiolic acid (CBDA), delta-9-tetrahydrocannabinol (Δ^9^-THC), 11-nor-9-carboxy-delta-9-THC (THC-COOH), delta-8-tetrahydrocannabinol (Δ^8^-THC), cannabicyclol (CBL), cannabinol (CBN), cannabichromene (CBC), cannabichromenic acid (CBCA), cannabidivarin (CBDV), cannabidivarinic acid (CBDVA), cannabigerol (CBG), cannabigerolic acid (CBGA), tetrahydrocannabivarin (THCV)) in methanol or acetonitrile (1.0 mg/mL) were Cerilliant^®^ certified reference materials, purchased from Merck-Sigma-Aldrich (St. Quentin Fallavier, France) and stored at −20 °C.

*Cannabis sativa* L. #138 with 4.5 % (w/dry plant) of CBD (CBDA+CBD) and 0.3 % THC (∆^9^-THCA+∆^9^-THC) is a female plant resulting from an agronomic test by the company LaFleur (Anger, France).

#### 2.1.2. Human Cell Lines and Correspondent Culture Media

Pancreatic human cell line was purchased from ATCC (LGC Standards, Molsheim, France). Cells were kept at 37 °C in humidified atmosphere containing 5 % (*v*/*v*) CO_2_ during their exponential growing phase and during incubation with investigated compounds. Confluency adherent cells were trypsinized and sub-cultured twice a week. Human pancreatic (AsPC-1, ATCC^®^ CRL-1682) cell lines were maintained in DMEM high glucose medium (Dominique Dutscher, 67172 Brumath, France, Cat No L0102-500), while human acute monocytic leukemia cell line (THP-1, ATCC^®^ TIB-202) was maintained in RPMI-1640 medium (ATCC^®^ 30-2001™, LGC Standards S.a.r.l., Molsheim, France), supplemented with 10% (*v*/*v*) heat-inactivated fetal bovine serum (FBS, Life Technologies, Paisley, UK, Cat No 10270-106) and 1% (*v*/*v*) penicillin–streptomycin (10^5^ units/mL and 10 mg/mL, Life Technologies, Paisley, UK, Cat No 15140-122).

### 2.2. Extraction Procedures from Cannabis #138

The stems, branches, leaves and trichomes were crushed with a manual “Grinder”. From this crushed plant material, three static maceration procedures were performed. For the first, 30.0906 ± 0.0001 g of crushed plant material were introduced into 200 mL of absolute ethanol, and then left to macerate for 10 days at −20 °C (#138-10D). For the second procedure, 3.2121 ± 0.0001 g of crushed plant material were introduced into 28 mL of absolute ethanol and left to macerate for 5 days at room temperature (20 ± 2 °C) (#138 RT-5D). For the last procedure, 3.2140 ± 0.0001 g of crushed plant material were introduced into 28 mL of absolute ethanol and then left to macerate for 2 days at room temperature (20 ± 2 °C) (#138 RT-2D). After maceration, the three liquids were filtered through a nylon cloth (pores 24 μm). The plant material was rinsed with a small volume of absolute ethanol (approximately 1 mL), and evaporated on a rotary evaporator (Rotavapor^®^ R-114 Büchi, Rungis, France) at 30 °C, 190 mbar, for 1.5 h. Finally, evaporation under a stream of nitrogen (Stuart^®^ SBH130D/3, France) was carried out for 12 h. The dry extracts obtained were stored in amber glass bottles protected from light at 4 °C.

### 2.3. Chemical Analyzes of Cannabis Extracts

#### 2.3.1. Quantification of Cannabinoids by HPLC

Liquid chromatography was performed on an Ultimate 3000 HPLC system (Thermo Fisher Scientific, Illkirch, France) combining a binary pump, a thermostatic well plate autosampler conditioned at 4 °C, a thermostatic column compartment conditioned at 30 °C, and a variable wavelength detector set at 228 nm. Chromatographic separation was achieved on a Raptor ARC-18 (Restek, 150 mm × 2.1 mm ID; particle size 2.7 µm, pore size: 90 Å). The mobile phase consisted of 25% solvent A (0.5 mM ammonium formate and 0.1% formic acid in water) and 75% solvent B (0.1% formic acid in acetonitrile). The flow rate was set at 0.4 mL/min, and the injection volume 20 µL. Standard calibration curves of CBD, CBDA, ∆^9^-THC, ∆^9^-THCA, ∆^8^-THC, CBL, CBN, CBC, CBCA, CBDV, CBDVA, CBG, CBGA and THCV were obtained by analyzing the successive dilutions of the standards solutions in acetonitrile (R² ≥ 0.9987). To analyze the PAs standards, a concentration sequence of 5, 10, 25, 30 and 40 µg/mL of the standards was obtained by diluting stock solutions by acetonitrile to obtain suitable sample solution in reverse phase chromatography. A mixture of each concentration was prepared and analyzed by HPLC. Identification of PAs peaks were performed based on the comparison of retention time of each standard. The linear range of the calibration curve was obtained and used for quantification.

#### 2.3.2. Identification of Terpenoids and Other Volatile Compounds by GC/MS

Extracts of *Cannabis sativa* L. #138-10D (after maceration at −20 °C for 10 days) were analyzed by GC/MS and adapted to our laboratory conditions. The extract (3.7 mg) was dissolved in ethanol to a final volume of 1000 µL and sonicated during 30 min. Then, 1 µL was analyzed with an Agilent 7890B gas chromatograph equipped with an Agilent 5977B inert MSD (Agilent Technologies). The gas chromatograph was fitted with a DB-Wax capillary column (60 m × 0.32 mm i.d. × 0.50 μm film thickness, J&W Scientific, Santa Clara, CA, USA) and helium was used as a carrier gas (1 mL/min at constant flow). The GC oven temperature was programmed without initial hold time at a rate of 2.7 °C/min from 70 °C to 235 °C (hold 10 min). The injector was set to 250 °C and used in splitless mode (25 psi for 0.50 min). The temperatures of the interface, MS ion source and quadrupole were 270 °C, 230 °C and 150 °C, respectively. The mass spectrometer was operated in electron impact ionization mode (EI, 70 eV) and the masses were scanned over a *m*/*z* range of 29–300 amu. Agilent MSD ChemStation software (G1701DA, Rev D.03.00) was used for instrument control and data processing. The mass spectra were compared with the NIST14 library reference spectral bank.

### 2.4. Cell Culture Methodology

Pure investigated compounds were initially dissolved in dimethyl sulfoxide (DMSO) in a concentrated stock solution. Further dilutions to the experimental concentrations applied on the cells were performed in RPMI-1640 or DMEM media prior to each experiment. Thus, the final concentration of DMSO on treated cells never reached higher than 0.5 % (*v*/*v*).

#### 2.4.1. Cancerous Organoids Formation in Liquid Pearls

In a first step, medium filling the pearls was prepared using DMEM ^high glucose^ complete medium supplemented by 1.2 % of methylcellulose solution (MC) at a ratio of 4 to 1. MC is a non-toxic product improving viscosity which, in turn, helps to form and maintain spheroids intact into the pearls. For spheroid generation, 150 µL of pearls medium were dispensed onto silica-coated 6-well plates; then, the AsPC-1 cells, at a density of 10^4^ cell/mL, were directly injected into each pearl and incubated in a humidified incubator at 37 °C and 5 % CO_2_. After 24 h, 50 µL of complete medium containing the different extracts (final concentration 80 µg/mL) were injected into each pearl and incubated for an additional 48 h. To evaluate cell viability, pearls were transferred into 96-well plates and incubated for 30 min with the injection of Calcein AM (Thermofisher, Illkirch, France) and propidium iodide (Life technology, Fischer scientific, Illkirch, France) according to the manufacturer instructions. Pearls containing cancerous organoids were finally analyzed on Celigo image cytometer (Cyntellect Inc, San Diego, CA, USA).

#### 2.4.2. Cell Cycle Analysis

Human pancreatic cells were seeded into 6-well plates at a density of 5 × 10^5^ cells/mL. For synchronization, the cells were incubated with DMEM-free serum for 48 h, then replaced with complete medium. After 24 h, the cells were exposed to 80 µg/mL of #138-10D extract then incubated for an additional 24 h.

### 2.5. TNF-α Secretion Assay

Fetal bovine serum (FBS) was obtained from Lonza BioWhittaker (Fisher Scientific, Illkirch, France). Penicillin–streptomycin was purchased from Cambrex Bio Science (St Beauzire, France). L-glutamine was obtained from Gibco (Invitrogen, Cergy Pontoise, France), and 96-wells culture plates were from Corning International (Avon, France). Lipopolysaccharide (LPS) from Salmonella abortus equi, RPMI 1640 was obtained from Sigma-Aldrich (Saint Quentin Fallavier, France). The human monocytic cell line THP-1 from ATCC was routinely maintained in RPMI 1640 culture medium supplemented with 10% (*v*/*v*) heat-inactivated FBS and penicillin–streptomycin (100 IU/mL and 1 μg/mL). Cells were incubated in 96-well culture plates for 6 h at 37 °C in a humidified 5% CO_2_-95% air atmosphere in the presence of increasing doses of tested compounds. The final concentration was 5 × 10^5^ cells/mL in a final volume of 250 μL per well. Lipopolysaccharide (LPS) from *Salmonella abortus equi* was used to induce TNF-α secretion (1 μg/mL). Compounds, dissolved in DMSO, were then diluted in culture medium and added to the cell cultures with a DMSO final concentration never exceeding 0.1% (*v*/*v*). Control samples always contained the same amount of DMSO (without drug) to exclude any interference of DMSO in cell responses. For the quantitative evaluation of secreted TNF-α the cells were stained.

### 2.6. Apoptosis Assay and Microcapillary Flow Cytometry Analysis

Cells were seeded in 96-well flat-bottom plates (Corning^®^ Costar^®^, Cat. No. CLS3596) in 0.1 mL at 10 × 10^3^ cells/well. PC-1 cells were left overnight to settle, while treatment of THP-1 cells started 2 h after seeding. Investigated compounds were added in a range of 6 concentrations. As controls, non-treated cells, cells treated with 0.5% DMSO, and cells treated with 50 µM Celastrol (Enzo Life Sciences, Cat. No. ALX-350-332-M025) were used. Plates were incubated for 15 min at 37 °C and supernatants were removed after an additional spinning cycle. Previously removed supernatants with non-adherent cells were returned to trypsinized cells and stained with Annexin V-FITC (ImmunoTools GmbH, Friesoythe, Germany, Cat No 31490013) and propidium iodide (PI, Miltenyl Biotec Inc, Auburn, AL, USA, Cat No 130-093-233) in a volume of 3 µL. Described trypsinization protocol was applied each time when AsPC-1 cells were prepared for flow cytometry analysis, unless stated otherwise. A minimum of 5 × 10^3^ cells were acquired per sample and analyzed on the InCyte software (Guava/Luminex, CA, USA). Apoptosis rates were assessed.

### 2.7. Mitochondrial Membrane Potential (ΔΨm) Measurement

Cells at a density of 20 × 10^3^ cells/well were seeded in 12-well plates, then were treated with 80 µg/mL of #138-10D extract for 48 h, and then were harvested. The mitochondrial integrity was then assessed by incubating the cells for 15 min at 37 °C with MitoPotential red solution (FlowCellectTM MitoPotential Red Kit, Millipore, France). Finally, a minimum of 2 × 10^3^ cells were acquired per sample and data computed with the Guava InCyte software (Guava/Luminex, CA, USA).

### 2.8. Measurement of Mitochondrial Reactive Oxygen Species (ROS)

In 12-well plates, 20 × 10^3^ cells/mL were plated and incubated overnight then treated with #138-10D extract at a concentration of 80 µg/mL for 6 h. After harvesting, the cells were resuspended and incubated with MitoStress Red dye (Flowcellect MitoSOX Kit, Millipore, France) for 30 min at 37 °C, followed by centrifugation and washing, and then resuspended in buffer assay, prior to capillary cytometry examination, as described earlier.

### 2.9. Caspases Activition Assay

The pancreatic cell line was plated in 12-well plates at a density of 20 × 10^3^ cells/mL. After 24 h, cells were incubated in the presence or absence of 80 µg/mL #138-10D extract for another 24 h. Cells were then collected and resuspended, then incubated for 1 h with the fluorescent markers, FITC- caspase-8 inhibitor IETD, FITC-caspase-9 inhibitor LEHD or FITC-caspase-3 inhibitor Z-VAD-FMK (CaspGLOW^TM^ Fluorescein Active Staining Kit, BioVision, CliniSciences, France). Cells were finally centrifuged, washed twice, and resuspend prior to analysis by capillary cytometry with the setting given by the dye manufacturer.

### 2.10. Statistical Analysis

Data, presented as bar graphs, were expressed as means ±S.E.M. of at least three independent experiments. Statistical evaluation was performed with the one-way ANOVA test followed by the post-hoc Bonferroni test using GraphPad Prism software (Prism version 5.04 for Windows, GraphPad Software, CA, USA).

## 3. Results

### 3.1. Quantification of Cannabinoids in the Three Different #138 Extracts by HPLC

An extraction method by static maceration was selected following the request of the LaFleur company. Static maceration is a traditional, very simple and relatively effective technique for the extraction of phytocannabinoids. In general, maceration without stirring is little used in the literature, since this method generally requires a relatively long extraction time. However, compared with dynamic maceration, it allows the use of a smaller amount of solvent and the ratio of crude plant extract/extraction solvent is 1/10. The ratio used for dynamic maceration is often greater, [3] for example 1/40 when using ethanol as the extraction solvent. Methanol and ethanol can be used as extraction solvent. They have similar physicochemical properties, but ethanol remains less toxic than methanol. Moreover, several publications have demonstrated better yields for extracting phytocannabinoids with ethanol [3,4]. Therefore, static maceration was chosen in this study and ethanol was used as the extraction solvent. In addition, the nature and the ratio of mass of solvent, and the duration and temperature of extraction are also very important parameters for static maceration, having a direct influence on the number and quantity of extracted molecules. Therefore, three extractions were performed, two using room temperature—one for 2 days (#138 RT-2D) and the other for 5 days (#138 RT-5D)—and one using −20 °C for 10 days (#138-10D). The identification and quantification of cannabinoids from the extracts were performed based on the retention time and the calibration curves of the corresponding authentic reference compounds. The amount of the fourteen most important cannabinoids determined for the three extracts, #138 RT-2D, #138 RT-2D and #138-10D, are presented in Table 1. Two cannabinoids, CBDA and CBD, were the major ones found in each extract, representing 56% and 36% of total cannabinoids, respectively. The amount of cannabinoids was similar for extracts #138 RT-2D and #138 RT-5D, indicating that, at room temperature, 2 days was long enough to extract the cannabinoids. However, the amount of almost all cannabinoids determined for #138-10D was about 20–25% higher than that for #138 RT-2D and #138 RT-5D. Thus, the extract #138-10D was selected for further studies.

### 3.2. Identification of Terpenoids and Additional Volatile Compounds in #138-10D Ethanolic Extract

As the *Cannabis sativa* L. plant has not only cannabinoids as crucial compounds, but also other compounds that may play an important role as synergistic and/or entourage compounds [5], it was crucial to identify such compounds present in the #138-10D extract. The main volatile compounds and their abundance are presented in Table 2. Results of this study are helpful in the evaluation of these compounds in mixture with cannabinoids, and later to assess their importance in medical treatment. The molecules previously described for their pro-apoptotic action that were present in substantial proportions in the #138-10D extract were, first of all, nerol, an acyclic monoterpenoid with 20.5% of peak area, then two sesquiterpenoids, guaiol (2.6%), and α-bisabolol (2.4%).

Nerol, a cis isomer monoterpene of geraniol, induces apoptosis in human colon cancer cells via the mitochondria-dependent pathway, as described by Menon and Gopalakrishnan [6]. Two other compounds, although minor in proportion, had previously been described as potential anticancer drugs:

(i) Guaiol, has been found in many medicinal plants and its roles in tumor suppression are still under investigation. Q. Yang and coll. [7], for instance, exhibited the significant role of Guaiol in inhibiting non-small cell lung cancer cell (NSCLC) growth. However, Guaiol is still a long way from being put into clinical practice. 

(ii) α-Bisabolol, a sesquiterpene alcohol, is described with a strong dose-dependent cytotoxic effect on human glioma cells, but it is non-toxic for normal glial cells [8]. α-Bisabolol induces a decrease in cell proliferation and viability in pancreatic cancer cell lines (KLM1, KP4, Panc1, MIA Paca2), but not in pancreatic epithelial cells (ACBRI515) [9].

It could be possible that Nerol, Guaiol and α-Bisabolol could account for an entourage effect in extract #138-10D even if, as we will see later, CBD comes out as the key actor in the observed effects on the apoptosis of pancreatic cancer cells.

### 3.3. Pancreatic Cancerous Organoids Generated in Liquid Pearls

Pancreatic cancerous organoids, generated for 24 h in liquid pearls, responded to #138-10D extract treatment, after 48 h incubation, in a manner equivalent to 80 µM Gemcitabine [10,11,12] used in therapy as a first-line treatment alone for pancreatic cancer (Figure 1). Thus, comparable results were observed with a concentration of 80 µg/mL of #138-10D extract, which represents an equivalent concentration of 40 µM CBD. It is interesting to note that Gemcitabine, one of the drugs that have been marketed and widely used in patients with pancreatic cancers, was only active on AsPC-1 cells at 80 µM in our experimental conditions.

### 3.4. Apoptosis Induction

To help differentiate, in a pharmacological manner, the three different extracts obtained by the three different extraction times and temperature conditions, our method of choice was to test them in dose-dependent manner for their induction of apoptosis. As presented in Figure 2, we chose to express the computed IC_50_ values in µM of CBD and not in mg/mL of dry extracts, as this later could be misleading by artificially highlighting one of the extracts. As CBD is prime candidate for proapoptotic activity [13], the equivalence in µM concentration of CBD contained in each extract thus generated a more accurate view of its importance.

The cytometry dot plots presented in Figure 3 illustrate that the majority of treated cells were labeled positive for annexin V, and this is true for any extract. The IC_50_ values obtained for n = 7 independent experimentations, with the different #138 extracts, were 26 µM for #138-RT-2D and #138-RT-5D, respectively, equal to the one of pure CBD (n = 3). For #138-10D extract, a slightly lower but close value of 20 µM (n = 7) was computed. Our results highlight a strong implication of CBD in the observed apoptosis induction.

### 3.5. Anti Inflammatory Activity

Tumor necrosis factor α (TNF-α) is a pro-inflammatory cytokine produced and secreted primarily by macrophages and monocytes in response to bacterial challenges or tumor burdens. The overproduction and secretion of TNF-α can have harmful effects strongly implicated in acute inflammation and in chronic inflammatory diseases. The objective was to assess the ability of extract #138-10D in parallel to CBD to inhibit the secretion of TNF-α. Activated THP-1 cells were pretreated and then exposed to a well-accepted inflammatory stimulus, such as LPS. The results indicate that after the stimulation of the THP-1 cells by LPS in the presence of each of the two treatments, a decrease in the fluorescence of the cells was obtained, with an inhibition of TNF-α secretion compared with the stimulated cells treated with vehicle DMSO only (Figure 4).

### 3.6. Cannabis Extract Induces Cell Cycle Arrest in Pancreatic Cancer Cells in the G0/G1 Phase

There is a strong argument for a link between the cell cycle, apoptosis, and mitochondrial mechanisms when cells are under stress, or following DNA destruction. The latter event, which can be induced by therapy, directly or indirectly causes activation of the genes essential for cell cycle regulation and apoptosis. Cell growth results from the progression of cells through the different phases of the cell cycle. For this purpose, the effect of extract #138-10D on the cell cycle was determined. An accumulation in the G0/G1 phase (74%) of AsPC-1 cells was observed compared with untreated cells (62%), while a logical decrease in the percentages of cells in the S and G2/M phase was observed. These results mean that extract #138-10D can already inhibit the growth of cancer cells within 24 h, and induce cell cycle arrest in the G1 phase (Figure 5) before inducing the apoptosis process 24 h later.

### 3.7. Mechanism of Action in Apoptosis Induction

#### 3.7.1. Generation of Mitochondrial ROS in Pancreatic Tumor Cells

Detection of mitochondrial superoxide ions was tested at the effective concentrations capable of inducing apoptosis. The data obtained by this test showed an absence of super-oxide ion generation in cells treated with CBD (data not shown). In contrast, cells exposed to the extract exhibited increased accumulation of intracellular superoxide ions through increased yellow fluorescence emission, when compared with untreated cells or vehicles (Figure 6A). In conclusion, all the data clearly show that the whole extract has a pro-oxidant character, while CBD alone has antioxidant potential. Cancer patients have been associated with high levels of oxidative stress markers such as ROS, nitric oxide (NO), oxidative DNA damage and lipid peroxides. In that aspect, the entourage molecules (e.g., Nerol or α-Bisabolol) help to equilibrate the homeostasis of radicals and anti-oxidants in organisms. As such, homeostasis is very important for normal metabolism, signal transduction and cellular function [14].

#### 3.7.2. The Effect of Cannabis Extracts on the Mitochondrial Integrity of Pancreatic Tumor Cells

Mitochondria play a central role in the life and death of cells, and are known to be the backbone of a wide range of diseases, including cancer. The unique structural and functional characteristics of mitochondria allow the selective targeting of drugs intended to modulate the function of this organelle for significant therapeutic gain. Loss of mitochondrial internal transmembrane potential is often associated with the early stages of apoptosis. Therefore, the effect of extract #138-10D on mitochondrial potential was analyzed here. The cancer cells were treated with the active concentrations of the extract (80 μg/mL) or of CBD (80 μM) for 24 h, followed by labeling with MitoPotential Red Kit dye. This kit is a dual-parameter assay that includes MitoSense Red, a cationic fluorescent dye that accumulates in mitochondria and responds to potential mitochondrial changes, and the DNA interlayer 7-Aminoactinomycin-D (7-AAD), a dye of dead cells impermeable to the membrane. Simultaneous use of the reagents allowed information on early and late apoptosis to be obtained in a single test. After analysis, an increase in the percentage of cells showing red fluorescence of dye in their nuclei was observed, compared with the untreated cells. For example, pancreatic tumor cells treated with extract #138-10D showed 73% 7-AAD-stained cells although negative for MitoSense Red, whereas for untreated cells this was only 12% (Figure 6B). This means that the treatment induced a change in the mitochondrial membrane potential and that the cell lost its membrane integrity.

#### 3.7.3. The Effect of Ethanolic Extract on Cell Caspases of Pancreatic Tumor Cells

Caspases are a family of cysteine proteases which attack and hydrolyze specific target proteins. The activation of caspases ensures that cellular components are degraded in a controlled manner, so their activations are considered an essential sign of programmed cell death. As shown in Figure 6C, the activation of caspases 3, 9 and 8 was very real when cells were exposed to a concentration of 80 μg/mL of extract #138-10D. Histograms demonstrated an increase in the populations labeled with the dye that specifically bound to active caspase. On the other hand, no fluorescence was observed in the untreated control cells.

## 4. Conclusions

A crucial first step in our study was to use of an original 3D culture model in liquid pearls. It helped to validate the efficacy of the extract in a test as close as possible to an in vivo animal model. Such a testing had to be performed, prior to any other experiment, to avoid any false positive responses often obtained in 2D flat biology. When in 2D, cancer cells are cultivated in monolayers, a long way from any metastatic structure (with low O_2_ inside, plus metabolite and anabolite gradients). Ethanolic CBD-rich cannabis extract induced, under our experimental conditions, more than 80% of apoptotic cell death when human pancreatic cells were exposed to 80 µg/mL for 48 h. The different extraction procedures (room temperature for 2 or 5 days, −20 °C for 10 days) did not significantly differ in their activity on cancer cell apoptosis. Chromatographic analyses of these extracts displayed content in CBD-A higher than CBD, though low in THC. The similarity of their cannabinoid contents could explain their similar apoptotic effects.

Additionally, we demonstrated here that CBD-rich cannabis not only activates the cell signaling cascade to lead to programmed cell death, but also causes cell cycle arrest in the G0/G1 phase. The ethanolic extract activates important factors involved in the regulation of the two processes mentioned above, and thus causes a change in the potential of mitochondrial membranes with a loss of mitochondrial integrity, with the consequence of great cellular stress.

## 5. Discussion

CBD-rich cannabis extract induced, in our experiment, a generation of ROS, thus explaining the cause of the loss of the potential of the mitochondria. We should mention that CBD does not induce ROS production, but still causes a change in mitochondrial membrane potential. We suggest that CBD could activate pro-apoptotic proteins linked to mitochondria independently of oxidative stress. It is also known that mitochondrial disruption causes activation of major markers of apoptosis, such as caspase-9 and -3. This could prepare their activation in cells treated with the extract. Our results also demonstrated an activation of caspase-8. This activation is known as the extrinsic pathway following the triggering of the death receptors. CBD-rich extract prepared by an ethanolic extraction procedure would therefore contain molecules that initiate apoptosis by additional mechanisms.

Finally, our analytic and biological results support the notion that soaking under a low temperature for a longer period is the best condition to produce an effective extract, in terms of quantity of bioactive molecules and anticancer activity. On the other hand, several companion compounds identified here will require prompt singular testing in the presence of rising concentrations of CBD. This will help to further decipher the mechanism of entourage effects observed with medicinal hemp extracts during the activation of death receptors and apoptotic cascades.

## Figures and Tables

**Figure 1 molecules-27-01214-f001:**
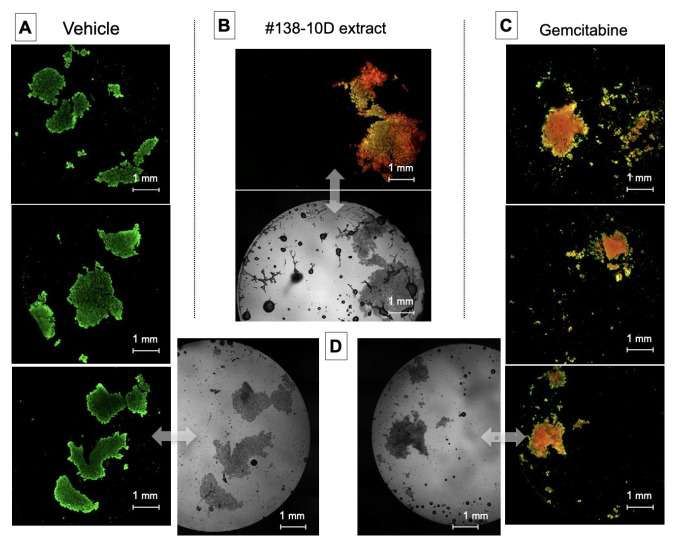
The effect of ethanolic extract of clone #138 on 3D-cultured cells’ viability. Pancreatic cancer cells seeded at 10^5^ cells/mL were injected and grown in liquid pearls at a volume of 150 μL for 24 h, then exposed to ethanolic extract #138-10D (80 µg/mL), vehicle or Gemcitabine (80 µM). After 24 h of incubation, the cells were labeled (Calcein-AM = green–living cells and propidium iodide = red–dead cells). The composite image of the two channels was analyzed by imaging cytometry.

**Figure 2 molecules-27-01214-f002:**
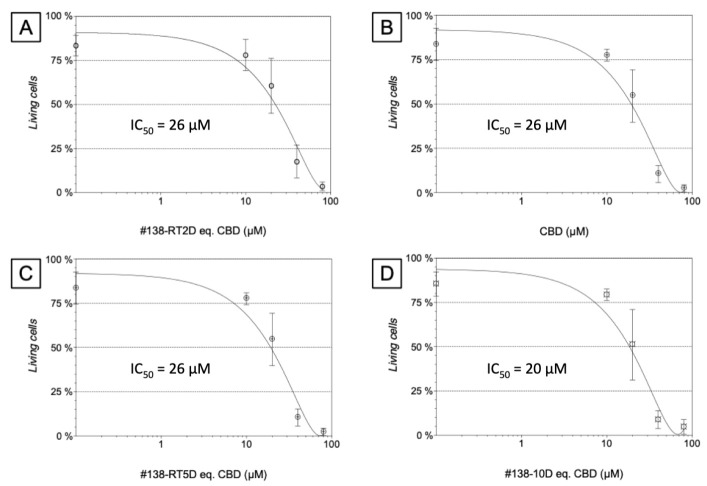
Effects of the different ethanolic extract on cellular apoptosis induction on the human pancreatic cancer cell line. The cells were treated with indicated concentrations for 48 h, and then induction of apoptosis was analyzed by flow cytometry using annexin-V and propidium iodide IP. Dose response estimates to find out the concentration that caused inhibition of 50% cell viability (IC_50_). Values are mean ±S.E.M of 7 independent experiments of the three #138 extracts with n = 3 for CBD alone. Computed EC_50_ values are given inside the graph for each treatment: (**A**) extract #138 RT-2D (**B**) pure CBD (**C**) extract #138 RT-5D (**D**) extract #138-10D.

**Figure 3 molecules-27-01214-f003:**
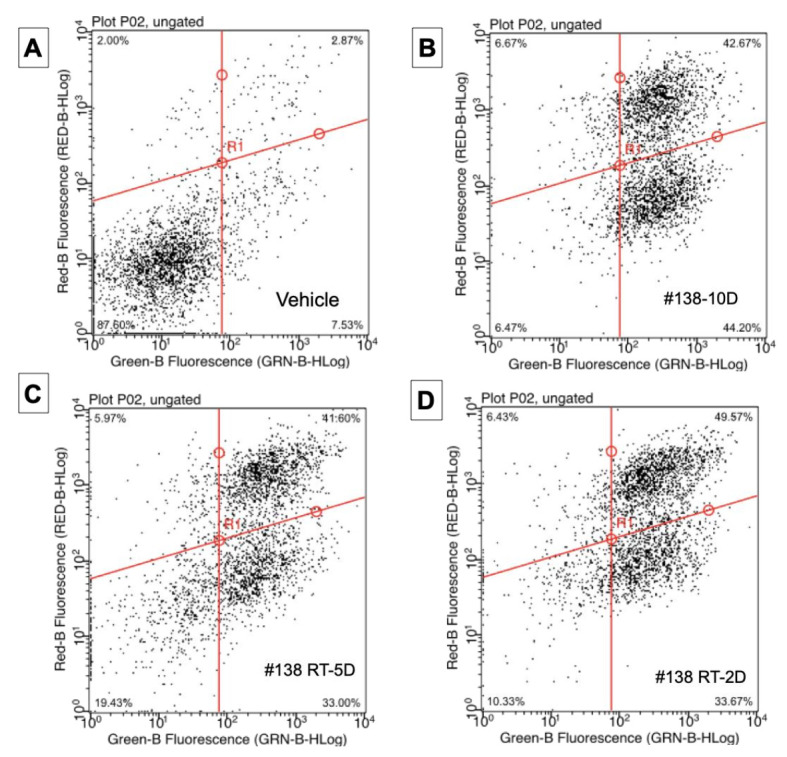
Apoptosis induced by ethanolic extracts on AsPC-1 pancreatic cancer cells. Cells were treated with 80 µg/mL of each extract for 24 h, then analyzed by capillary cytometry (annexin-V and PI staining). Dot plots (**A**) vehicle, (**B**) extract #138-10D (**C**) extract #138 RT-5D (**D**) extract #138-2D, with % of living cells (bottom left annexin-V ⊖, IP ⊖), early and late apoptosis (right column: annexin-v ⊕, early IP⊖ or late IP ⊕) and cells in necrosis (top left: annexin-v ⊖, IP ⊕).

**Figure 4 molecules-27-01214-f004:**
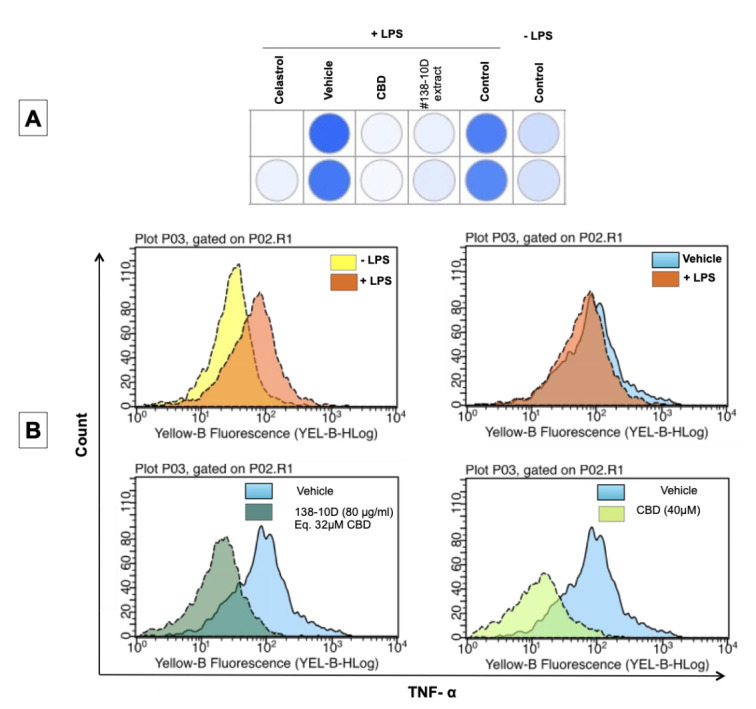
Anti-inflammatory activity assay. THP-1 monocyte cells were treated as indicated for 1 h, then were stimulated with LPS for 2.5 h followed by incubation with a TNF-α detection antibody (TNF-α secretion assay). The expression of membrane bound TNF-α was detected by flow cytometry. (**A**) Analysis shows the level of TNF-α release in the presence or absence of the LPS stimulator and treatments (dark blue = high level release, gray color = inhibition level of TNF-α). (**B**) Representative histograms of fluorescence intensity versus cell number.

**Figure 5 molecules-27-01214-f005:**
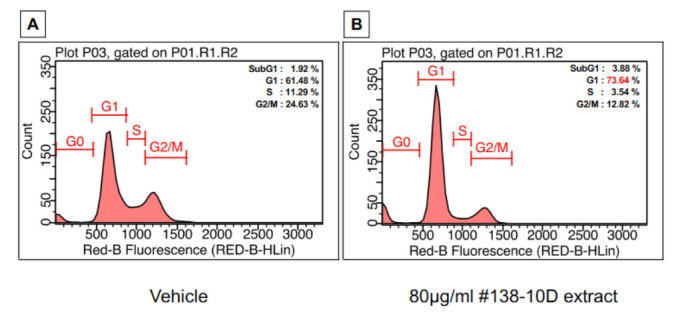
Cell cycle progression analysis. Cells were exposed to the extract at 80 µg/mL and incubated for 24 h. Cell cycle distribution was determined by a capillary cytometry detection assay. DNA content histograms for treated (**A**) and untreated (**B**) cells show the percentage of cells during SubG1, G1, S and G2/M.

**Figure 6 molecules-27-01214-f006:**
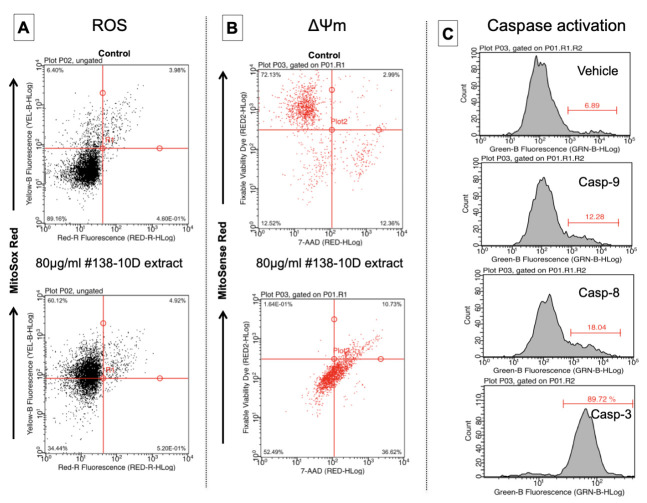
Detection of mitochondrial ROS generation, Mitochondrial potential, and Caspase activation in pancreatic tumor cells. Cells were exposed to 80 µg/mL of #138-10D then incubated with specific dyes as described in M and M. (**A**) Analyzed data after incubation with a Flowcellect MitoSOX Kit. Dot plots show the percentage of unlabeled cells (Top left—ROS^−^) or labeled cells (Bottom left—Ros^+^) that showed an increase in yellow fluorescence. (**B**) Analyzed data after staining with Mitopotential Red Kit. Quadrant gates show downward shift in fluorescence for treated cells undergoing a change in their mitochondrial potential (lost their integrity). (**C**) Histograms show the percentage of positively stained cells by FITC green fluorescence (a shift to the right represents activation of caspase-3, 9 or 8).

**Table 1 molecules-27-01214-t001:** Identification and quantification of cannabinoids for 3 different ethanolic extraction procedures of #138. Mean values for two independent injections. With RT = room temperature; 2D = two days; 5D = five days; −10D = −20 °C for 10 days. Data generated by LCA, INRAE, Toulouse.

Extraction Procedure	#138 RT-2D	#138 RT-5D	#138 -10D
Cannabinoids	µg/mg *	µg/mg *	µg/mg *
CBDVA	1.67	1.49	1.54
CBDV	6.10	4.58	6.01
CBDA	252.09 (55%)	253.37 (56%)	312.93 (55%)
CBGA	nd	nd	nd
CBG	nd	nd	nd
CBD	162.21 (36%)	158.91 (36%)	201.45 (36%)
THCV	nd	nd	nd
CBN	0.89	0.94	1.03
∆^9^-THC	9.26	8.53	12.01
∆^8^-THC	nd	nd	nd
CBL	0.98	1.55	1.26
CBC	6.63	6.45	8.47
∆^9^-THCA	3.85	3.79	4.81
CBCA	10.02	8.99	12.42
**THC total (∆^9^-THCA + ∆^9^-THC)**	**13.11**	**12.32**	**16.82**
**CBD total (CBDA + CBD)**	**414.30**	**412.28**	**514.38**

Legend: nd = not detected, * mg of dry extract.

**Table 2 molecules-27-01214-t002:** Identification of terpenoids and other volatile compounds in the #138-10D ethanolic. Data generated by Twistaroma, Illkirch.

CAS	Common Name	Chemical Family	Validation	Delta RI	RT (min)	Peak Area	% of Total Peak Area
628-97-7	Ethyl hexadecanoate	Fatty acid esters	MS, RI	4	28.25	3 724 514	34.75%
106-25-2	Nerol	Acyclic monoterpenoids	MS, RI	23	24.77	2 193 368	20.47%
104-76-7	2-Ethyl hexanol	Fatty alcohols	MS, RI	0	18.97	632 378	5.90%
102-62-5	Glycerol 1,2-diacetate *	Diacylglycerols	MS		25.97	396 506	3.70%
489-86-1	Guaiol	Sesquiterpenoids	MS, RI	36	26.50	264 182	2.47%
515-69-5	α-Bisabolol	Sesquiterpenoids	MS, RI	6	27.90	252 436	2.36%
1000150-76-3	(+)-Borneol		MS		22.14	224 289	2.09%
507-70-0	Borneol	Bicyclic monoterpenoids	MS, RI	1	22.14	223 007	2.08%
1209-71-8	Ɣ-Eudesmolɣ	Cycloeudesmane sesquiterpenoids	MS		27.00	208 364	1.94%
112-05-0	Nonanoic acid	Medium-chain fatty acids	MS, RI	29	27.12	186 551	1.74%
124-19-6	1-nonanal	Aldehydes	MS, RI	4	17.90	169 331	1.58%
6813-21-4	Selina-3,7(11)-diene	terpene	MS, RI	23	23.42	164 038	1.53%
142-62-1	Hexanoic acid	Medium-chain fatty acids	MS, RI	2	23.57	151 905	1.42%
473-15-4	b-Eudesmol	terpene	MS, RI	29	28.49	140 380	1.31%
124-63-0	Methane-sulfonyl chloride	Sulfonyl chlorides	MS, RI	17	14.55	109 761	1.02%
79-09-4	Propanoic acid	Carboxylic acids	MS, RI	8	19.75	109 213	1.02%
22122-36-7	3-methyl-5H-furan-2-one	Butenolides	MS, RI	13	22.52	106 067	0.99%
112-31-2	Decanal	Aldehydes	MS, RI	1	19.41	95 975	0.90%
473-16-5	α-Eudesmol	terpene	MS, RI	17	28.32	92 419	0.86%
58319-06-5	(1R,5R)-2-Methyl-5-((R)-6-methylhept-5-en-2-yl) bicyclo[3.1.0] hex-2-ene	Sesquiterpenoids	MS, RI	26	20.75	89 568	0.84%
111-71-7	Heptanal	Aldehydes	MS, RI	25	13.99	86 317	0.81%
116-09-6	Acetol	Ketones	MS, RI	5	16.66	83 925	0.78%
87-44-5	β-caryophyllene	Sesquiterpenoids	MS, RI	6	21.18	74 950	0.70%
124-06-1	Ethyl tetradecanoate	Fatty acid esters	MS, RI	34	25.81	67 140	0.63%
124-13-0	Octanal	Aldehydes	MS, RI	4	16.22	66 858	0.62%
1632-73-1	Fenchol	Bicyclic monoterpenoids	MS, RI	0	20.57	66 076	0.62%
79-33-4	lactic acid	#N/A	MS, RI	11	17.12	59 564	0.56%
75-05-8	Acetonitrile *	Nitriles	MS, RI	20	11.62	57 611	0.54%
106-32-1	Ethyl octanoate	Fatty acid esters	MS, RI	2	18.46	44 641	0.42%
3913-81-3	trans-2-decenal	Aldehydes	MS, RI	1	21.42	42 356	0.40%
10482-56-1	L- α-Terpineol	Menthane monoterpenoids	MS		21.96	40 320	0.38%
3796-70-1	(E)-geranyl acetone	Acyclic monoterpenoids	MS, RI	2	23.81	35 766	0.33%
54446-78-5	1-(2-Butoxyethoxy)ethanol	Hemiacetals	MS, RI	46	23.06	35 330	0.33%
78-70-6	Linalool	Acyclic monoterpenoids	MS, RI	23	19.80	32 137	0.30%
577-27-5	ledol	terpene	MS		22.74	31 179	0.29%
18794-84-8	trans-β-farnesene	Sesquiterpenoids	MS, RI	29	21.49	29 727	0.28%
111-87-5	1-Octanol	Fatty alcohols	MS, RI	1	19.91	29 529	0.28%
460-01-5	Cosmene		MS		15.27	26 963	0.25%
123-35-3	β-myrcene	Acyclic monoterpenoids	MS, RI	5	14.16	26 489	0.25%
67-68-5	Dimethyl sulfoxide	Sulfoxides	MS, RI	13	20.34	25 218	0.24%
22106-07-6	Pyrimido[1,6-a] indole, 1,2,3,4-tetrahydro-2,5-dimethyl-1,2,3,4-tetrahydropyrimido (3,4-a) indole]	indole alkaloids	MS		24.87	18 526	0.17%
17301-30-3	Undecane, 3,8-dimethyl-	Branched alkanes	MS		18.12	18 504	0.17%
626-89-1	4-methyl-1-pentanol	Primary alcohols	MS, RI	23	16.88	17 900	0.17%

* Artefact resulting from GC/MS pollution.

## Data Availability

Not applicable.

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
