# Peer review of "Cannabis sativa Extract Induces Apoptosis in Human Pancreatic 3D Cancer Models: Importance of Major Antioxidant Molecules Present Therein"

_molecules, 2022, doi:10.3390/molecules27041214_

Round 1

Reviewer 1 Report

The article describes the use of cannabinoids as anti-cancer therapeutics. The authors compared 3 different extraction techniques to obtain the highest concentration of cannabinoids in the extract. They also determined other compounds, which could play a synergetic role in the process. Several biological assays were performed, from which the pancreatic cancerous organoids generated in liquid pearls is an original approach. The reviewer thinks that the study is nicely conducted and scientifically sound but should be backed up with more literature.  I added some comments to improve the manuscript.

  1. In the introduction a review is missing of all the work that is already performed in the last years about the topic. For example which extraction methods other authors used until now on the cannabis plant. Are the extraction methods that you used already performed in other applications. Is there a reason why exactly these extraction methods. Some literature support on the methods of qualification and quantification of your compounds (description of difficulties in the determination of the compounds found from former literature and why you choose this method of analysis), if the analysis has been always straightforward, mention in materials and methods the literature on which your measurements is based, since your method is probably not invented from scratch.
  2. The statements you use in the introduction are not supported with any literature. I am sure that after each sentence a suitable reference can be found, and multiple for lines 45and 48.
  3. After the description of the former literature, there should be emphasized what you did in this research, that has not been done before or what has been done differently.
  4. For sections 2.4 (with subsections) to 2.9. Refere to the literature where you took the methods from. If these are standard procedures, the descritpions can be shortened, or describe which adaptions were made to the standard method.
  5. you have a section 2.2.1 but not section 2.2 (line 90), make it 2.2. And below you have a subsection 2.2.1.1 but no section 2.2.1.2. Don't make subsecstions where there is only 1 item.
  6. I also see in section 2.2 to 2.9 that lots of items are still containing brand names and place of origin, such as ATCC (LGC Standards, Molsheim, France). These descriptions can be all moved to section 2.1. Make 2 subsections if needed, for example 2.1.1 chemicals and reagents and 2.2.2 materials (or whatever section head would fit better)
  7. lines 264-288 should be moved to the introduction. This type of explanation is exactly what I was missing there.
  8. line 285 needs a reference. What about different ethanol:water ratios? or hot water extraction? Or was it already proven that cannabinoids are not stable in such solvents? From where the decsion of 2, 5 and 10 days
  9. lines 299-301, 304-320, 392-402 is also for introduction
  10. line 266: can you define 'as much as possible': quantity or also types of cannabinoids, because it is known that with different extractions often different types of compounds can be extracted. Is there one type of compound that is of higher interest than others?
  11. Table 2: describe abbreviations. I would remove the row 'followed by 20 other compounds'. If you write in the title 'determination of major terpenoids', it will be obvious that the sum of the peaks will not be 100%. I do not like that your identification of compounds is not backed up with literature since you only identified compounds with a nist library, which is prone to errors. 

Author Response

All remarks where taking into account.

The responses as the changes induced in the final text will appear as « comments" in the revised word file. 

Reviewer 2 Report

Only minor comments:

L 123.  Please described in more details the quantitation of cannabinoids.  Was a single calibration curve used for all quantitations?  How precise is such calibration.

L 124. Volatil must be volatile

Table 2.  I suggest replacing "surface" with "area"

Author Response

(The authors gave the same response as above.)

Reviewer 3 Report

Abstract

-It focused only  on methods of extraction without clarifying the effect of extract on the apoptotic markers. please consider this point

Introduction 

-It is not enough and has clear shortage in detailed data about the previous studies related to the present work.

-There is clear shortage in the provided references (only one reference) rather than whole the manuscript.

Methods

-They are well described however there was no any references supporting them

Results and discussion 

-results were enough and well described but there was not enough interpretation based on other studies

-Discussion very little and did not reflect the impact of the results

Conclusion

-Very long. must be rewritten in a concised form. many parts of it should in incorporated in the discussion and Abstract (in brief). 

whole manuscript needs moderate english editing improvement 

Author Response

(The authors gave the same response as above.)

Reviewer 4 Report

The study design and experiments in the article titled "Cannabis sativa extract induces apoptosis in human pancreatic 3D cancer models: importance of major antioxidant molecules present therein" are of scientific value and significance. But the presentations and writings are not up to the mark. So, I have some review suggestions to improve the quality of the article:

1. Background information in the “Introduction” section is very limited. Moreover, no background reference for the previous study regarding anticancer activity and/or cancer cell death and/or molecular mechanism (apoptosis or any other) and the role of cannabinoid(s) have been mentioned. An “Introduction” section containing background information with appropriate references supporting the purpose of the study is a prerequisite. My suggestion is to restructure this section with sufficient reference information in accordance with the purpose of the study.

2. There are many spelling mistakes and few grammatical errors. Some are- e.g., somewhere mL and somewhere ml, somewhere Aspc-1 and somewhere AsPC-1 etc.

3. English should be carefully corrected and reviewed.

4. A brief explanation in favor of the selection of maceration conditions (i.e., temperature, solvent etc.) is necessary. Solvents of standard reference materials are different (i.e., methanol/acetonitrile) from solvent for maceration. Is there any solvent effect during maceration? Mention with reference.

5. Section 3.3 should be transferred to the “Materials and Method” section.

6. The presentations in the sub-sections of “Result” section is disorganized. Figures are not appropriate in sense of representation and figure legends are not self-explanatory; because each individual sub-figure (i.e., A, B, C. D etc.) has not been explained in the figure legend. Authors should include these. And also correct the image content and quality rather than the representation of raw figures.

6. Follow the numberings in subsections of the “Result” section, i.e., 3.1, 3.2, 3.3, 3.4, 3.5, 3.7 etc…

7. Result data should be described and compared in the subsections of “Result” section which is absent in few subsections (e.g., subsection “Generation of mitochondrial ROS in pancreatic tumor cells”) and in my opinion “Discussion” should be a separate section possessing insights correlating the found data and the justification of the purpose of the study.

Author Response

(The authors gave the same response as above.)

Round 2

Reviewer 1 Report

I have a feeling not of my comments are completely answered

I personally still think that the introduction can use more references, so readers know what has been done before. 

I'm not sure if other authors performed in earlier research already Identification of terpenoids and other volatile compounds from hemp extracts, I'm still missing a comparison from your data with these (in case it exist, doesn't matter if it's a different way of extraction) since no standards were used to confirm the results - maybe i did not express it ok in my last review.

Author Response

Please, find here attached, the text to be added beginning of §3.1 giving 6 references to previous work.  
